# Selected Matrix Metalloproteinases (MMP-2, MMP-7) and Their Inhibitor (TIMP-2) in Adult and Pediatric Cancer

**DOI:** 10.3390/diagnostics10080547

**Published:** 2020-07-31

**Authors:** Aleksandra Kaczorowska, Natalia Miękus, Joanna Stefanowicz, Elżbieta Adamkiewicz-Drożyńska

**Affiliations:** 1Department of Pediatrics, Hematology and Oncology, Faculty of Medicine, Medical University of Gdańsk, 7 Dębinki Street, 80-952 Gdańsk, Poland; miekus@gumed.edu.pl (A.K.); edrozynska@gumed.edu.pl (E.A.-D.); 2University Clinical Centre, 7 Debinki Street, 80-952 Gdansk, Poland; 3Department of Pharmaceutical Chemistry, Faculty of Pharmacy, Medical University of Gdańsk, al. Gen. J. Hallera 107, 80-416 Gdańsk, Poland; natalia.miekus-purwin@gumed.edu.pl; 4Faculty of Health Sciences, Medical University of Gdańsk, Maria Sklodowska-Curie Street 3a, 80-210 Gdańsk, Poland

**Keywords:** tumor microenvironment, metalloproteinases, tissue inhibitors of metalloproteinases, pediatric cancer

## Abstract

The tumor microenvironment (TME) consists of numerous biologically relevant elements. One of the most important components of the TME is the extracellular matrix (ECM). The compounds of the ECM create a network that provides structural and biochemical support to surrounding cells. The most important substances involved in the regulation of the ECM degradation process are matrix metalloproteinases (MMPs) and their endogenous inhibitors (tissue inhibitors of metalloproteinases, TIMPs). The disruption of the physiological balance between MMP activation and deactivation could lead to progression of various diseases such as cardiovascular disease, cancer, fibrosis arthritis, chronic tissue ulcers, pathologies of the nervous system (such as stroke and Alzheimer’s disease), periodontitis, and atheroma. MMP-TIMP imbalance results in matrix proteolysis associated with various pathological processes such as tumor invasion. The present review discusses the involvement of two MMPs, MMP-2 and MMP-7, in cancer pathogenesis. These two MMPs have been proven in several studies, conducted mostly on adults, to make an important contribution to cancer development and progression. In the current review, several studies that indicate the importance of MMP-TIMP balance determination for the pediatric population are also highlighted. The authors of this review believe that carrying out biochemical and clinical studies focused on metalloproteinases and their inhibitors in tumors in children will be of great relevance for future patient diagnosis, determination of a prognosis, and monitoring of therapy.

## 1. Introduction

Cancers are the second most common cause of death in adults globally. In children, globally, more than 300,000 diagnoses of cancer are made each year. The biology of the tumor in children and adults is extremely different, but the urge to understand mechanisms leading to cancer development in both groups is understandable. In recent years, scientists have focused particular attention on the tumor microenvironment (TME). The TME surrounding the cancer cell includes the extracellular matrix (ECM), blood vessels, fibroblasts, immune system cells, and numerous substances secreted by these cells. It has been proven that interactions between cancer cells and the TME may contribute to the development, progression, and formation of cancer metastases. The TME has also a significant impact on the effectiveness of treatment. One of the most important components of the TME is the ECM [1]. In healthy tissue the ECM consists mainly of collagens, fibronectin, elastin, laminin, entactin, and proteoglicans. These create a network that provides structural and biochemical support to surrounding cells. Every organ has a unique composition of the ECM to serve a particular tissue-specific purpose. The balance between degradation and synthesis of ECM components is the key factor to maintain homeostasis and the proper functioning of the tissue. The TME-associated ECM is fundamentally different from that of the healthy tissue stroma. In cancer, a disturbed balance between ECM synthesis and secretion, and changed expression of matrix-remodeling enzymes lead to abnormal ECM dynamics. The tumor-derived ECM is biochemically distinct in its composition and has greater rigidity than the normal ECM [1]. Researchers have noted that the ECM plays a crucial role in tumor proliferation by providing tumor cells with sustaining growth signals, contributing to avoid growth suppressors, and eventually resisting cell death. One of the key steps leading to the growth and spread of cancer is the degradation of the ECM [2]. The most important substances involved in the regulation of the ECM degradation process are matrix metalloproteinases (MMPs) and their inhibitors (tissue inhibitors of metalloproteinases, TIMPs). The present review discusses the recent findings related to MMPs, particularly MMP-2 and MMP-7, and their involvement in various cancers. In addition, the dual role of TIMP-2 (as the main inhibitor of MMP-2) in cancer is demonstrated. This review is subdivided into general biochemical and clinical sections. The main goals of this thesis are to highlight the necessity for new trials to particularly focus on the involvement of selected MMPs and TIMP-2 in pediatric tumor pathogenesis and to emphasize their potential usability in assessing patient prognoses.

## 2. MMP—Characteristic and Classification

In the early 1960s, the metalloproteinases (MMPs), also called matrixins, became known as the key enzymes responsible for degradation of the extracellular matrix and non-matrix proteins. These form a family of more than 23 distinct multidomain zinc-dependent and calcium-dependent endopeptidases (numbered 1 to 3, 7 to 17, 19 to 21, and 23 to 28 for historical reasons). Their expression is controlled by inflammatory cytokines, growth factors, and hormones such as EMMPRIN (extracellular matrix metalloproteinase inducer). They also have endogenous inhibitors (TIMPs), cell-anchored protein RECK (reversion-inducing cysteine-rich protein with kazal motifs), and α2-macroglobulin, the major MMP inhibitor in serum [3,4,5]. MMPs are not only involved in the regulation of cell expression, morphogenesis, wound healing, and bone resorption, but also in apoptotic processes, angiogenesis, and metastasis. An additional important role of MMPs is the control of extracellular reservoirs of inflammatory cytokines and chemokines because MMP-9 can release tumor necrosis factor alpha (TNF-α), and MMP-2 and MT1-MMP take part in the release of IL-1β [5]. Therefore, MMPs are described as being both protective of and detrimental to tissues, depending on the biological process involved. In addition, the disruption of the physiological balance between MMP activation and deactivation could lead to progression of various diseases such as cardiovascular disease, cancer, fibrosis arthritis, chronic tissue ulcers, pathologies of the nervous system (such as stroke and Alzheimer’s disease), periodontitis, and atheroma [6,7].

The most general classification of MMPs divides them into collagenases, gelatinases, stromelysins, matrilysins, membrane-type MMPs (MT-MMPs), and a heterogenous group containing MMP-12 (Macrophage metallo-elastase), enamelysin (MMP-20), and epilysin (MMP-28) (Table 1) [8].

Most MMPs have the same basic domains comprising propeptidic, catalytic, hinge, and hemopexin-like domains. The ∼80-residue zymogenic pro-peptide domain is situated at the amino-terminal (N-terminal), whereas, at the carboxy-terminal (C-terminal), ∼165-residue zinc- and calcium-dependent catalytic and ∼200-residue hemopexin-like domains are situated [7,9,10]. Further insertions include fibronectin type-II-related within the catalytic domain (characteristic for MMP-2 and MMP-9), a collagen type-V-like insert; a vitronectin-like insertion domain; a cysteine or proline-rich and interleukin-1 receptor-like domain; an immunoglobulin-like domain; a glycosyl phosphatidylinositol linkage signal; a type-I or type-II trans-membrane domain; or a cytoplasmic tail. In addition, the MMP structure includes a membrane anchor from the MT-MMP subfamily [7]. MMPs are synthesized and secreted as transmembrane proenzymes. The removal of an N-terminal pro-peptide, which keeps the enzyme in latent form, is necessary for their proteolytic activation. MMPs do not only overlap in their domain similarities, but also in substrate specificity, spatial and temporal location, and in their ability to interact with other enzymes, such as disintegrin and metalloproteinases (ADAMs) to cleave upstream of hydrophobic residues [11].

Collagenases are the enzymes that break down four types of collagen (I, II, III, and IV). Interstitial collagen fibrils resist degradation by most proteinase enzymes. MMP-1 and MMP-13 are more prone to cleave the collagen types II and III, respectively. In contrast, degradation of type I collagen by MMP-8 is around three times stronger than that by MMP-1 or MMP-13. The initialization of cleavage of collagen by collagenases allows fibrillar collagen fragments to become susceptible to further degradation by various MMPs such as MMP-2, MMP-3, and MMP-9. Collagen is not the only target of collagenase enzymatic action; other notable targets are gelatin, L-selectin, interleukin-1, entactin, ovostatin, MMP-2, MMP-9, proteoglycans, aggrecan, fibronectin, and plasminogen [12,13]. MMP-8 is a key collagenolytic MMP in the ethiopathogenesis of periodontal disease [14,15,16] and contributes to the development of type II diabetes [17]. Collagen type I fragments (gelatin) are prone to fibronectin domains in MMP-2 and MMP-9, thus achieving the cleavage of gelatin. Other known substates for both gelatinases are elastin, pro-transforming growth factor β (pro-TGF-β), pro-TNF-α; solely for MMP-2, tenascin and fibronectin; and solely for MMP-9 plasminogen, entactin and vitronectin, among others [5]. MMP-2 and MMP-9 have been studied in the pathological stages of the organism in which the leukocyte migration from the circulation into the tissue during inflammation was observed. These studies were mainly performed in a murine model of experimental autoimmune encephalomyelitis (EAE) that has similarity to the human disease multiple sclerosis [18].

The stromelysin subgroup comprises MMP-3, MMP-10, and MMP-11. MMP-3 is an enzyme capable of degrading laminin, fibronectin, gelatins of type I, III, IV, and V, collagens, and cartilage proteoglycans. Another member of this subfamily, MMP-11, unlike almost all other MMPs, is activated before secretion by Golgi-associated furin-like proteases. In addition, this stromelysin is unique in that its active form cannot hydrolyze classical MMPs substrates, and it can only cleave bioactive mediators β-casein, α2-macroglobulin, and serine proteinase inhibitors. Among the stromelysins, little is known about the function of the last member of the group, MMP-10, but there is increasing interest in the importance of this MMP in kidney disease, for example. Moreover, its expression has been confirmed: It is expressed by macrophages in numerous tissues after injury. Considering the identity between MMP-10 and MMP-3 in their sequence of the catalytic domains and active sites, this pair has been indicated to be the most homologous of all MMPs [19,20,21,22].

MMP-7 is a matrilysin. It is one of the smallest (28 kDa) known members of this MMP family. Although the gene coding MMP-7 is within a well-conserved MMP cluster on chromosome 11q21–23, this particular MMP has the unique feature of being expressed by the exocrine and mucosal epithelium that is not subject to injury or inflammation. Moreover, MMP-7 and MMP-26 do not include the hinge region and hemopexin domain, which is required for interactions with other MMPs and TIMP. Additionally, MMP-26 is associated with the intracellular milieu, which is also unusual for MMPs because most are secreted into the ECM [23,24,25].

MT-MMPs are a group that comprises six molecules. They are unique among MMPs because they possess the transmembrane domain at the C-terminus (MMP-14, MMP-15, MMP-16, MMP-24) or, in the case of MMP-17 and MMP-25, the glycosylphosphatidylinositol (GP) anchor attaches them to the plasma membrane. MT1-MMP (MMP14) was the first member of the group to be identified and has the widest substrate specificity. Because MT1-MMP degrades fibrillar collagens, including types I, II, and III, it has a significant role in the degradation of the most abundant ECM components. MMP-15 and MMP-16 are also able to degrade collagens (types I and III, respectively), while the other members of the transmembrane MMPs do not cleave the fibrillar collagen. Several MT-MMPs have been shown to activate proMMP-2 on the cell surface and can stimulate the release of ECM-associated growth factors, such as vascular endothelial growth factor (VEGF) or TGFβ [9,26,27].

Finally, we discuss the heterogenous subgroup of the MMP family. MMP-12 is connected to the activation of macrophages, which further leads to the amplification of the inflammatory cascade, and could be linked to the pathogenesis of several lung diseases such as chronic bronchitis, pulmonary emphysema, and asthma. In addition, the other MMP-12 mechanism involves the cleavage of N-cadherin and the release of β-catenin, which is linked to several life-threatening diseases such as cancer, aging, and osteoporosis or degenerative disorders [28]. MMP-20, also called enamelysin, digests primarily amelogenin. It is expressed by ameloblasts and odontoblasts. Enamelysin differs from other MMPs because its pattern of expression is highly restricted and this MMP is considered a tooth-specific MMP [29,30]. MMP-28, also named epilysin, is the newest and, apparently, last member of the mammalian MMP family. Its mRNA is expressed at high levels in the mesenchymal stem cells, testes, lungs, heart, GI tract, and in wounded epidermis in human tissues. Its main substrates are casein and neural cell adhesion molecule (NCAM) [31,32].

## 3. TIMPs—Characteristic and Classification

There are four homologous members of the TIMP family with a similar secondary structure of six loops stabilized by six disulfide bonds (created by 12 highly conserved cysteine residues). The TIMPs’ molecular weight is roughly 21 kDa. In their structure each of the domain N- and C-terminals has three loops. The net charge of the C-terminal domain varies between TIMP members: TIMP-1 and TIMP-3 are positively charged, and TIMP-2 and TIMP-4 are negatively charged. The N-terminal subdomain of each TIMP molecule contains the inhibitory activity for the degenerative potential of the MMPs; however, the role of the TIMP C-terminal domain in MMP inhibition is yet not fully understood [33]. The role of TIMPs in ECM turnover can be described as the inhibition of all known MMPs with varied efficacy. TIMP-1 has strong inhibiting properties against MMP-9 but poorly inhibits MT1-MMP, MT3-MMP, MT5-MMP, and MMP-19. TIMP-2 preferably cleaves MMP-2 and forms a complex composed of homodimers of TIMP-2 and MMP-2 (Figure 1) or TIMP-2-pro-MMP-2-MT1-MMP, which leads to the activation of pro-MMP-2.

TIMP-2 can also inhibit other MMP members. TIMP-4 is able to form this complex, but using a slightly different mechanism. Additionally, TIMP-3 also has the ability of forming such a complex. The normal tissue interactions between MMPs and TIMPs are essential for tissue remodeling. Localization of TIMPs is relevant for controlling the enzyme function. TIMP-3 has two unique attributes amongst the TIMPs: it binds to the ECM through interaction with heparan sulfate and other sulphated proteoglycans, and it is the primary TIMP to cleave most ADAMs [34,35]. TIMP-2 is distinguished amongst the TIMP family members because it can directly suppress the proliferation of endothelial cells. 

The factors that control the gene expression of TIMPs in the tissue cells (other than inflammatory and tumor cells) are not well known, whereas there are a large number of factors known to stimulate the gene expression of TIMPs, such as cytokines (IL-1; IL-6; IL-11), retinoic acids, LPS, hormones, steroids, oncogenic transformation; viral infection, growth factors, and chemical and physical stimuli. TGF-β, in contrast, is known to suppress TIMP-2 while stimulating TIMP-1 [36]. 

MMP-TIMP imbalance results in matrix proteolysis associated with various pathological processes, such as tumor invasion. The potential of MMPs and TIMPs as new therapeutic agents for cancer has been also discussed [37,38,39]. The invasion by cancer cells of surrounding tissues is strongly connected with interactions that occur between invasive cancer cells, cells producing various MMPs (such as endothelial, stromal, and neutrophils cells and macrophages), and cells producing TIMPs (such as lymphocytes, monocytes, macrophages, and mast cells) [40]. It has been noted that TIMPs have both the ability to stimulate and suppress tumor growth. For example, TIMP-2 demonstrates its antiangiogenic activity by inhibiting MMPs, however, TIMP-2 also binds to the endothelial cell surface to suppress VEGF-A-stimulated endothelial mitogenesis [41,42]. In contrast, the association between TIMP-2 and cell migration has been proven. TIMP-2 binds to overexpressed MT1-MMP which further stimulates the MEK/ERK pathway and cell migration, and leads to the promotion of the invasive function of MT1-MMP [43]. Such findings have been further investigated and several studies proved that elevated concentrations of TIMP-2 positively correlate with an unfavorable prognosis in cancer patients.

## 4. The Involvement of Metalloproteinases and Their Inhibitors in Cancer

The process of cancer invasion and metastasis includes several steps, among which degradation of ECM is essential. As MMPs and their inhibitors play a key role in this process, several interesting studies have been carried out to reveal their role in cancer. These studies have been mainly conducted on the adult population, and only a few have been conducted on the pediatric population. For the purpose of this review the detailed characteristics of MMP-2, MMP-7, and TIMP-2 (as the main inhibitor of MMP-2) are presented. According to the available data, MMP-2 and MMP-7 are among the most important metalloproteinases in the pathogenesis of tumor formation. Because of significant differences between adult and pediatric malignancies, these two groups will be discussed separately.

### 4.1. Adult Population—The Involvement of MMPs and TIMP

#### 4.1.1. MMP-2 Involvement in Cancer

MMP-2 is a metalloproteinase that has a proven role in the pathogenesis of cancer. Multiple studies have revealed an increased expression of MMP-2 in TME including cancer of the colorectum, ovaries, breast, prostate, bladder, lung, and pancreas, in addition to primary skin melanoma and central nervous system (CNS) malignancies (Table 2). 

Langenskiold et al. examined tumor samples, tumor-free bowel samples, and plasma obtained from 72 patients with colorectal carcinoma. The results showed increased concentration of MMP-2 (among others) in tumor tissue compared with tumor-free tissue. Lymph node status correlated with MMP-2 plasma level, which was distinctly elevated only in patients with lymph node metastasis. In conclusion, MMP-2 plasma level may possibly be used as a predictor in colorectal malignancy [44]. A similar observation was made by Hilska et al., who examined tumor samples obtained from 351 patients with primary colon or rectal cancer. High expression of MMP-2 in cancer cells was associated with decreased survival of colon cancer patients [45]. 

In ovarian cancer, the role of MMP-2 is controversial—certain study results indicated an association between MMP-2 overexpression and prognosis, while others did not reach such conclusions [46,64,65]. It is certain, however, that ovarian carcinomas showed significantly higher levels of MMP-2 than benign and borderline tumors [66]. Peringy and colleagues examined tissue samples from ovarian tumors and peritoneal implants in 100 patients. Analysis showed that only MMP-2 overexpression by cancer cells in peritoneal implants was associated with a significant risk of death by progression of the disease. The results led to the conclusion that MMP-2 overexpression by cancer cells in peritoneal implants, but not in primary ovarian cancer, is predictive of ovarian cancer prognosis and more likely reflects the presence of highly aggressive clones of cancer cells [47].

MMP-2 is also associated with the development, aggressiveness, and overall survival rate in breast cancer. Nearly 30 years ago, Davies et al. analyzed samples collected from 11 patients with benign breast tumors and 32 with breast cancer, and they demonstrated a clear relationship between the production of type IV collagenases and malignant breast disease. The ratio of active to total MMP-2 was increased in high-grade tumors, although the total amount of enzyme was not increased [48]. Several years later, Iwata et al. achieved consistent results, additionally suggesting that activation of pro-MMP-2 may be an indicator of lymph node metastasis in breast cancer [49]. Talvensaari-Mattila et al. explored the role of MMP-2 as a prognostic factor, and proved (by examining tumor samples taken from 453 patients) that decreased concentration of MMP-2 can serve as a marker for favorable prognosis in breast cancer [50]. 

Morgia et al. focused their research on prostate cancer by collecting plasma from 40 patients with prostate cancer (20 with organ-confined carcinoma and 20 with metastatic disease), 20 with benign prostate hyperplasia, and 20 healthy males. The level of MMP-2 (among other MMPs) was increased in samples obtained from patients with metastatic prostate cancer compared to other groups, and decreased markedly after treatment started [51].

Regarding bladder carcinoma, excretion of MMP-2 in urine has been associated with a high grade and high stage of the disease [52]. Elevated levels of MMP-2 in urine were mainly observed in patients with invasive tumors. Authors also mentioned that, for bladder cancer patients, the elevated levels of MMP-2 were found in bladder tissue samples [53,63]. Nonetheless, the non-invasiveness of urine collection makes this biological fluid favorable for biochemical biomarker research. In addition, the data indicated that biological fluid concentration of MMP-2 correlates with the levels found in tissue samples. Therefore, urinary levels of MMP-2 could serve as a potential supportive biomarker for the diagnosis and monitoring of the patients with bladder cancer [52]. 

Interesting results were also found from examining cerebrospinal fluid (CSF) in patients with malignancies of the CNS compared to those without tumors. All patients with positive CSF cytologies had activated MMP-2 [54]. These results were confirmed by Sawaya et al., who presented proof that expression of MMP-2 is significantly upregulated in malignant gliomas and correlated with progression of human gliomas [55]. 

MMP-2 was examined in several other cancers, such as lung adenocarcioma, pancreas cancer, and primary skin melanoma [56,57,58]. In all of these studies, results showed poorer prognosis in patients with increased MMP-2 level. Therefore, MMP-2 could serve as a promising biomarker for every step of oncological care: diagnosis, monitoring of the treatment effects, determination of prognosis, and follow-up of patients. 

#### 4.1.2. TIMP-2 Dual Role in Cancer

The results relating to TIMP-2 are less equivocal than those of MMP-2. It is known that in low concentration TIMP-2 activates MMP-2, and that high levels of TIMP-2 inhibit MMP-2 activity. As a result, it was expected that a high level of TIMP-2—by inhibiting MMP-2 and its procancerous activity—will result in inhibition of cell growth and tumor spread, improving patient prognosis. However, several studies have yielded conflicting results regarding the anti-tumor effect of TIMP-2. This is probably due to the fact that not all functions of the TIMP proteins are currently known. Scientists suspect that the positive association between TIMP-2 and the proliferation marker topoisomerase II α may suggest its mitogenic effect [67]. The anti-cancer nature of TIMP-2 is supported by the results of studies in which the downregulation of TIMP-2 was observed, among others, in malignant gliomas or in the progression of prostate cancer [68,69]. For lung cancer cell lines, the scientific data revealed that expression of TIMP induced death of lung cancer cells and overexpression of p53. Pulukuri et al. examined prostate cancer cell lines and confirmed that in prostate tumor the silenced TIMP-2 gene was associated with cancer progression during the invasive and metastatic stages of the disease [69]. On the other hand, patients with bladder or ovarian cancer have been shown to have higher levels of TIMP-2 [63,66]. Nakopoulou et al. conducted a study of 136 breast cancer tumor samples and concluded that patients with increased tumor sizes often demonstrated negative TIMP-2 expression. Furthermore, higher levels of TIMP-2 were detected in most cases of low-grade patients, and these patients also had better survival. However, scientists have demonstrated a positive connection between TIMP-2 and bcl-2, thus indicating the potential anti-apoptotic function of TIMP-2 [67].

#### 4.1.3. MMP-2/TIMP-2 Imbalance in Cancer

The concentrations of MMP-2 and TIMP-2 measured separately in cancer patients are notable, and the sensitive relationship between MMP-2 and TIMP-2 is also of great interest among scientists and clinicians. Subsequent studies have shown that an imbalance between MMP-2 and TIMP-2 concentrations in tissues can determine tumor invasion and metastasis. Particular attention has been paid to the imbalance between MMP-2 and TIMP-2 in hepatocellular carcinoma (HCC), and cervical, bladder, breast, and oral cancers (Table 2). 

Giannelli et al. performed a study on 40 patients diagnosed with HCC, in which serum and tumor tissue levels of MMP-2 and TIMP-2 were measured at the time of diagnosis, and the clinical outcome was then followed over a two-year period. While serum and tissue levels of MMP-2 were not statistically different in patients with or without metastasis, TIMP-2 levels in these samples were significantly elevated in HCC patients without metastasis. Moreover, 90% of patients with increased TIMP-2 levels were still alive after two years, whereas only 30% with low levels of TIMP-2 survived. These results suggest that an MMP-2/TIMP-2 imbalance (and particularly TIMP-2 levels) could become a valuable prognostic factor in patients with HCC [59].

The correlation between MMP-2/TIMP-2 concentration and disease outcome has also been an object of interest in cervical carcinoma. Davidson and co-workers found MMP-2 mRNA almost exclusively in cancer cells, and found that it was lacking in preinvasive dysplastic lesions and healthy tissue. Additionally, MMP-2 levels strongly correlated with higher stage and poor survival. TIMP-2 was detected in both invasive tumor cells and benign epithelial cells in two control cervices. Cervix samples from a control group were also found to be negative for MMP-2, which can indicate a protective role of TIMP-2 in non-cancerous cells; however, as the control group was limited, these data are of less importance. In conclusion, researchers suggested that the presence of mRNA for both MMP-2 and TIMP-2 is related and associated with poor survival [60]. Another study conducted by Talvensaari-Mattila et al. also focused on cervical carcinoma. They tested serum obtained from 12 cervical carcinoma patients and 27 healthy volunteers. The medium levels of TIMP-2 and MMP-2/TIMP-2 complex in serum were higher in healthy women than in cases with a malignant tumor [61]. The imbalance between MMP-2 and TIMP-2 has been also connected to the increased metastatic and invasive phenotype of lymph-node-positive breast cancer [70].

In a study on oral cancer tumor tissue, Shretsha et al. concluded that the rate of MMP-2/TIMP-2 complex expression is more precise for characterization of MMP-2 activity [62]. This observation was supported by data obtained by Kanayama et al., who focused their study on expression of MMP-2 and TIMP-2 in bladder cancer. The expression of both MMP-2 and TIMP-2 was higher in invasive tumors, and was strongly associated with poorer survival [63].

#### 4.1.4. MMP-7 Involvement in Cancer

In addition to MMP-2 and TIMP-2, MMP-7 is also relevant for tumor pathogenesis. Researchers have proved MMP-7 overexpression in many tumors, including esophageal, gastric, colorectal, pancreas, ovarian, prostate, lung, and bladder cancers (Table 3).

Miao et al. carried out a meta-analysis of 14 clinical cohort studies (number of tumor samples included in the study, i.e., 935, was meaningful), discussing MMP-7 expression in esophageal cancer. The results were promising—they showed an obvious correlation between overexpression of MMP-7 and higher TNM (T-tumor, N-nodus, M-metastases) stages, higher invasive grade, and presence of lymph nodes metastasis [71]. The expression of MMP-7 is barely detected in healthy gastric mucosa. Okayama et al. proved that MMP-7, among others, was a significant biomarker for prediction of lymph node metastasis in primary gastric cancer. It was also strongly correlated with depth of invasion and stage of the disease [72]. Similar results were presented in the research of Yamashita et al.—MMP-7 levels were higher in gastric cancer with vascular invasion compared to a group without invasion [73]. Maurel et al. analyzed serum obtained from 120 patients with colorectal cancer. The key finding of this study was that MMP-7 is an independent prognostic factor for survival in advanced colorectal cancer, and is possibly even more accurate than lactate dehydrogenase. High MMP-7 serum level also correlated with liver involvement [74]. Studies conducted by Polistena et al. led to similar conclusions—data obtained with the use of an immunohistochemistry-based method revealed higher expression of MMP-7 in advanced cancer than in non-metastatic disease. In addition, a higher level of MMP-7 in the serum of stage III/IV patients was noticeable compared to patients in stage I/II of the disease [75].

Kuhlmann et al. concentrated their research on patients with pancreatic diseases. At present, carbohydrate antigen 19-9 (Ca 19-9) is one of a small number of biomarkers that can aid doctors in making correct diagnoses of pancreatic diseases. Because differentiating between periampullary carcinoma and chronic pancreatitis with an inflammatory mass can be challenging, the researchers sought a new biomarker to help in choosing the right treatment strategy in patients with a periampullary mass. Researchers analyzed levels of MMP-7 in plasma and pancreatic juice of 94 patients (63 with pancreatic neoplasm, 31 with chronic pancreatitis). The results indicated that median plasma MMP-7 levels were significantly higher in carcinoma than in cases with chronic pancreatitis, but the MMP-7 level in plasma alone is not sufficient as a positive predictive value. However, combined MMP-7 and Ca 19-9 levels in plasma gave a positive predictive value of 100%. Regarding pancreatic juice, MMP-7 levels were higher in carcinoma, but not significantly so [78]. Pancreatic cancer was also of interest in a study by Jones et al., who examined, among others, MMP-7 levels in tumor tissue. The MMP-7 level was significantly higher in pancreatic cancer than in the normal pancreas and was related with reduced survival [77]. These observations were supported by a study by Yamamoto et al. [76].

MMP-7 may also be a candidate marker for ovarian cancer detection and a possible target for therapeutic intervention. Tanimoto et al. analyzed samples obtained from 44 patients with ovarian tumors and 10 samples from patients with healthy ovaries. Results showed overexpression of MMP-7 in both high-grade ovarian cancer and low-grade malignant potential ovarian tumors, and an absence of MMP-7 expression in healthy ovaries. Lymph node metastases were histopatologically proven in five cases, and all of these cases showed overexpression of MMP-7 [79].

In addition to the digestive tract and ovaries, the importance of MMP-7 for prostate cancer has also been proven. Conor et al. created a rodent model that mimics the osteoblastic and osteolytic changes associated with human metastatic prostate cancer. They proved that MMP-7 in the tumor-bone microenvironment was an important mediator of prostate cancer-induced osteolysis [82].

An increased level of MMP-7 was also found in non-small cell lung cancer. Liu et al. tested tumor tissue obtained from 147 patients and in 51.6% of cases overexpression of MMP-7 was revealed. Additionally, overall survival was significantly lower in patients with overexpression of MMP-7 [80].

Szarvas et al. focused their research on patients with bladder cancer. They gathered 179 individuals (160 with cancer, 19 healthy individuals), and tested tumor samples in addition to serum. The study showed overexpression of MMP-7 in tissue samples of metastatic bladder cancer, whereas in cases with localized disease the expression of MMP-7 was normal. High tissue expression levels were accompanied by elevated serum MMP-7 concentration. Additionally, high tissue and serum MMP-7 levels were connected with poorer prognosis and higher risk of distant metastasis [81].

### 4.2. Pediatric Cancer and MMP-2, MMP-7, and TIMP-2 Involvement

Pediatric malignancies are an extremely heterogenous group of cancers. They concern a considerably smaller number of patients than in the adult population and there are have been fewer clinical trials involving children compared to adult populations (Table 4).

This is probably a reason for the smaller number of studies examining the role of MMPs and TIMPs in tumor pathogenesis in contrast to the adult population. All of the studies discussed below, except for those concerning leukemia, were performed on tumor tissue/cancer cell-lines, and none were focused on non-invasively obtained samples such as serum or urine.

Although leukemias are the most frequent malignancies in childhood, among which acute lymphoblastic leukemia (ALL) is the most common, relatively few studies have investigated the role of MMPs and TIMPs in leukemia cell invasion. At present it is known that there is no expression of MMP-2 in healthy bone marrow mononuclear cells. It is also acknowledged that leukemic cells express MMP-2, however, the precise role of MMP expression in ALL is still not clear [100]. Moreover, there are clear differences between MMP-2 expression in adult and childhood ALL, which were highlighted in the research of Kuittinen et al. In this study, bone marrow aspirate smears obtained from 20 adults and 55 children were investigated. The results showed that, in adults, 65% of cases were positive for MMP-2 expression, and in children only 12.7% of cases were positive. In adults, there was also a statistically significant correlation between overexpression of MMP-2 and the existence of extramedullary infiltrates, while in children there no such correlation was observed. In pediatric cases, positive MMP-2 expression has been related to the high-risk tumor group and T-cell immunophenotype [83]. Most recent studies have focused on the connection between MMP-2 and MMP-7 genotypes and the risk of childhood leukemia. Researchers from Taiwan concluded that although MMP-2 promoter genotypes play a minor role in the assessment of the risk of developing leukemia, the MMP-7 A-181G genotype may serve as a predictive biomarker for childhood ALL [101,102].

Acute myelogenous leukemia (AML) is another type of leukemia that concerns the pediatric population (although much less often than ALL). The crucial role of MMP-2 in AML was the subject of a study by Sawicki et al. They prepared a human leukemic cultured KG-1 cell line and examined the role of MMP-2 in in vitro invasion by leukemic cells, by using recombinant human TIMP (rhTIMP) and anti-MMP-2-antibody. The results revealed that after inhibiting the activity of MMP-2 the invasiveness of KG-1 cells decreased by 76% and 51%, respectively [84]. Similar results, concerning MMP-7, were delivered by Lynch et al. In this research two leukemic cell lines—K562 (chronic myelogenous leukemia blasts; more aggressive) and HL-60 (acute promyelocytic leukemia blasts; less aggressive)—were analyzed for expression of several MMPs and their inhibitors. The expression of MMP-2 and TIMP-2 was similar in both cell lines, but only K562 cells expressed MMP-7. Additionally, by inhibiting MMP-7, a 40% reduction of invasiveness was revealed. This study emphasized that MMP-7 may play an important role in leukemia cell invasion [99].

Lymphadenopathy is a common symptom in children, and most cases are non-cancerous. Differentiation between reactive and malignant lymphadenopathy is often problematic, therefore, the urge to find new biomarkers to help diagnose these patients is understandable. Stolarska et al. gathered 34 tissue samples of childhood lymphomas and reactive lymph nodes. Among lymphomas, Hodgkin lymphoma (HL) was the most common (13 samples), then Burkitt and T-cell lymphoma (3 samples each), and the least common was anaplastic large cell lymphoma. The TIMP-2 expression was examined, and the researchers observed TIMP-2 expression exclusively in lymphomas. The disadvantage of this work was the lack of MMP-2 expression results [97]. Pennanen et al. performed two studies on the role of MMPs and TIMPs in lymphomas, although the population was not strictly pediatric. In the first study, the clinicopathological role of TIMP-1 and TIMP-2 in HL was investigated by examining tumor samples obtained from 68 patients. In the results, scientists stressed that TIMP-1 expression could promote the growth of HL, but TIMP-2 expression correlated with systemic symptoms, which are usually associated with more advanced stages of disease [98]. Four years later, the same team published a manuscript in which MMP-2/TIMP-2 complex levels were measured in plasma obtained from 126 patients with lymphomas (HL = 31, non-Hodgkin lymphoma (NHL) = 95) and 44 healthy controls. Patients with newly diagnosed, active lymphoma had higher levels of MMP-2/TIMP-2 complex and lower levels of TIMP-2, compared to healthy individuals. Additionally, the patients with highest plasma levels of MMP-2/TIMP-2 complex had a greater chance of relapse of the disease [89].

The most common solid tumors in children are CNS tumors. Gu et al. collected 45 samples of pediatric gliomas and 20 normal brain tissue samples, and then analyzed them for extracellular matrix metalloproteinase inducer (EMMPRIN) and MMP-2 expression. The results relating to anaplastic astrocytoma and glioblastoma revealed distinctly higher levels of expression of both substances in anaplastic astrocytoma and glioblastoma than in healthy brain and low-grade astrocytoma tissue. Patients with overexpression of EMMPRIN and MMP-2 also had lower survival rates [85].

Another common malignancy in childhood is neuroblastoma, a tumor arising from the primitive neuroepithelial cells of the neural crest. In 1998, Ara et al. published a study in which they examined tumor tissues of 31 patients with neuroblastoma for MMP-2, MMP-9, and TIMP-2 expression using immunohistochemical staining with specific antibodies. Increased MMP-2 expression, in addition to decreased TIMP-2 expression, had a significant association with more advanced stages, and therefore, could possibly serve as a prognostic factor in patients with neuroblastoma [90]. Two years later, similar results were obtained (also by Ara et al.) by examining 25 neuroblastoma tumor samples for the expression of MMP-2, MMP-9, and TIMP-2 using reverse-transcription polymerase chain reaction. Higher levels of MMP-2 mRNA and the ratio of MMP-2/TIMP-2 mRNA were observed in advanced stages and in patients who died from the progression of the disease. The expression of MMP-9 and TIMP-2 were not significantly associated with clinical stages and prognosis [91]. The connection between MMP-2 and MMP-9 expression and the stage of the disease has been confirmed in other studies [92,103]. Ribatti and co-workers studied neuroblastoma tumor samples using immunohistochemical staining to determine the expression of MMP-2 and MMP-9. Results revealed that the expression of MMP-2 and MMP-9, among others, was increased in advanced stages of the disease [93].

Several studies have also focused on sarcomas, because these tumors are characterized by high levels of, among other MMPs, MMP-2. Rhabdomyosarcoma (RMS), a malignant tumor of skeletal muscle origin, divides into two major histological subtypes: Embryonal rhabdomyosarcoma (ERMS) and alveolar rhabdomyosarcoma (ARMS), which is much more aggressive subtype. Onisto et al. tried to determine whether the aggressiveness of ARMS cells may depend on differential expression of specific MMPs, their inhibitors, and VEGF. Four ARMS, three ERMS and one undifferentiated sarcoma (UDS) cell lines were examined. The analysis revealed that MMP-2 was overexpressed in three of the four ARMS lines, in contrast to only one ERMS cell line. TIMP-2 levels were not significantly different among the cell lines. In addition, the ARMS cell line with increased expression of MMP-2 was more invasive compared to the ERMS cell line with low levels of MMP-2 [86]. Consistent results were obtained by Diomedi-Camassei et al., who examined 33 human RMS samples (12 ARMS, 21 ERMS). ARMS showed stronger MMP-2 expression compared with the ERMS type. To summarize, due to their capacity to promote tumor growth and invasiveness, MMPs may be a factor that increases the aggressiveness of ARMS [87]. Osteosarcoma is the most common histological form of primary bone cancer. It mostly affects teenagers and young adults. Bjorland and colleagues prepared a study on osteosarcoma cell lines (OS). As expected, the most invasive cell line contained the highest amounts of MMP-2 [88]. Roomi et al. took a different approach to the subject—they examined the effects of cytokines, mitogens, inducers, and inhibitors on MMP-2 expression in OS and RMS. The results showed that all of the studied compounds caused upregulation of OS and RMS MMP-2 secretion, and inhibitors caused downregulations. These data may suggest that application of these agents could be useful in the treatment of sarcomas [104].

Expression of MMP-2 and TIMP-2 was also studied in retinoblastoma (RB), the most common primary malignant intraocular cancer in children. Adithi et al. used a specific biochemical test based on antibodies to examine 62 tumors for, among others, MMP-2 and TIMP-2. Among all of the tumor samples, MMP-2 was expressed in 66% of them and TIMP-2 in 53%. MMP-2 and TIMP-2 were overexpressed in invasive tumors (84.8% and 75%, respectively). Researchers also observed higher expression of MMP-2 in poorly differentiated RB compared with moderately/well-differentiated RB. There was no connection found between MMP-2 and TIMP-2 expression and clinical outcome or laterality [94]. Furthermore, Webb et al. examined MMP-2 (and MMP-9) as one of the targets for therapy of RB metastases. Two RB cell lines were prepared: one with high metastatic potential (Y79) and one with low metastatic potential (Weri-1). Then, inhibitors of MMP-2 and MMP-9 were used. Results showed reduction in Y79 cell line migration ability, and a significant decrease of viability in Weri-1 cell line. Moreover, in both cell-lines, inhibition of MMP-2 reduced secretion of transforming growth factor β-1 (TGF-β1) [95].

One of the first significant studies examining the expression of MMP-2 and TIMP-2 in pediatric tumor cells was that undertaken by de Veas et al. in 1994. They prepared four lines of cell cultures: two of neuroblastoma cells (007 and WAC-2, which was more aggressive), one of osteosarcoma cells (U-2OS), and one of rhabdomyosarcoma (A-204). With the exception of the A-204 cell line, MMP-2 levels were significantly increased compared to TIMP-2 levels. In the more aggressive WAC-2 neuroblastoma line, MMP-2 expression was twice as high as that in the less aggressive 007 cells. However, the highest level of MMP-2 was determined for the U-2OS cell line. Regarding TIMP-2, it was expressed in all four of the cell lines, but there was a significant difference between specific cell lines. The expression of TIMP-2 was almost twice as high in the U-2OS cell line as in the A-204 cell line. In neuroblastoma, TIMP-2 was overexpressed in the less aggressive 007 line compared with expression in the WAC-2 cell line [96].

## 5. Summary

Based on the studies cited above, it can be clearly stated that the levels of the selected metalloproteinases and their inhibitors are disturbed in both adult and child cancers. However, there is a notable difference in the number of patients tested. The population of pediatric cancer patients is significantly smaller and more heterogeneous compared to the population of adult cancer patients. Hence, conclusions about the role of metalloproteinases and their inhibitors in the development of childhood cancer are often drawn from a statistically unsatisfactory number of patients. Scientists continue to seek new tools for early detection of the disease, biomarkers in determination of prognoses, and novel approaches to treatment. MMPs and TIMPs appear to be promising candidates for novel therapy approaches, but more clinical studies, particularly of the pediatric population, need to be undertaken. It would be also of great value to encourage studies based on non-invasively obtained samples, as these can serve as a useful tool to assess disease remission and to predict disease relapse. Future studies should implement knowledge relating to the MMP-TIMP imbalance in pediatric cancer, thus accelerating cancer diagnosis and determination of a prognosis, and improving monitoring of patient therapy. In combination, this could lead to administration of specialized medicines for children diagnosed with cancer.

## Figures and Tables

**Figure 1 diagnostics-10-00547-f001:**
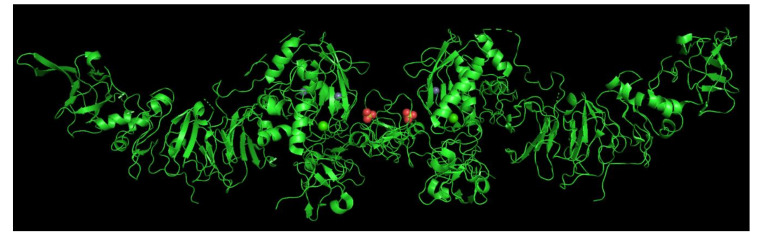
Matrix metalloproteinase (MMP)-2-(PDB ID:1QIB) tissue inhibitors of metalloproteinase (TIMP)-2 (PDB ID: 1BR9) binding. The Ca^2+^ ion is essential for homodimerization of MMP-2. The complex was visualized with the PyMOL 2.3.4 system. Spheres definition: red: SO4^-^; green: Ca^2+^; purple: Zn^2+^

**Table 1 diagnostics-10-00547-t001:** Human matrix metalloproteinase family members.

Collagenases		Gelatinases		Stromelysins	
MMP-1	Collagenase-1 Interstitial collagenase	MMP-2	Gelatinase A 72-kDa type IV collagenase	MMP-3	Stromelysin-1
MMP-8	Collagenase-2 Neutrophil collagenase	MMP-9	Gelatinase B	MMP-10	Stromelysin-2
MMP-13	Collagenase-3			MMP-11	Stromelysin-3
**Matrilysins**		**Membrane-Type MMPs**		**Other MMPs**	
MMP-7	Matrilysin	MMP-14	Membrane type-1 MMP	MMP-12	Macrophage metallo-elastase
MMP-26	Matrilysin-2/endometase	MMP-15	Membrane type-2 MMP	MMP-20	Enamelysin
		MMP-16	Membrane type-3 MMP	MMP-28	Epilysin
		MMP-17	Membrane type-4 MMP		
		MMP-24	Membrane type-5 MMP		
		MMP-25	Membrane type-6 MMP		

**Table 2 diagnostics-10-00547-t002:** Summary of selected studies of MMP-2 and TIMP-2 role in adult cancers.

MMP/TIMP	Type of the Cancer	Type of the Sample	Main Findings	References
**MMP-2**	Colorectal cancer	Tumor tissue Plasma	Increased concentration of MMP-2 in tumor tissue, compared with tumor-free tissue MMP-2 plasma level was distinctly elevated in patients with lymph node metastasis compared with those without	Langenskiold et al. [44]
		Tumor tissue	High expression of MMP-2 in cancer cells was associated with decreased survival of colon cancer patients	Hilska et al. [45]
	Ovarian cancer	Tumor tissue	Significant relationship between activated MMP-2 and invasiveness, metastasis, disease progression; MMP-2 as a potential marker of prognosis	Wu et al. [46]
		Ovarian tumors /peritoneal implants	MMP-2 overexpression by cancer cells in peritoneal implants and not in the primary ovarian cancer is predictive of ovarian cancer prognosis	Peringy et al. [47]
	Breast cancer	Tumor tissue	The ratio of active to total MMP-2 was increased in high-grade tumors	Davies et al. [48]
		Tumor tissue	Activation of pro-MMP-2 may be an indicator of lymph node metastasis in breast cancer	Iwata et al. [49]
		Tumor tissue	Decreased MMP-2 concentration could serve as a marker for favorable prognosis in breast cancer	Talvensaari-Mattila et al. [50]
	Prostate cancer	Plasma	The level of MMP-2 increased in patients with metastatic prostate cancer compared to BPH/healthy patients	Morgia et al. [51]
	Bladder cancer	Urine	Urinary excretion of MMP-2 is associated with higher stage and grade; it may indicate tumor progression and predict relapse of the disease	Gerhards et al. [52]
		Tumor tissue	Levels of active MMP-2 increased with tumor grade and invasiveness	Davies et al. [53]
	Central nervous system malignancies	CSF	All patients with positive CSF cytologies had activated MMP-2	Friedberg et al. [54]
		Tumor tissue	Expression of MMP-2 significantly upregulated in malignant gliomas and correlated with progression of human gliomas	Sawaya et al. [55]
	Lung adenocarcinoma	Tumor tissue	MMP-2 positive patients had poorer prognosis	Kodate et al. [56]
	Pancreas cancer	Tumor tissue	MMP-2 plays an important role in tumor cell invasion and leads to progression of the disease	Ellenrieder et al [57].
	Primary skin melanoma	Tumor tissue	MMP-2 positive patients had poorer prognosis	Vaisanen et al. [58]
**MMP-2/TIMP-2**	HCC	Serum + tumor tissue	MMP-2/TIMP-2 imbalance (and particularly TIMP-2 levels), could become a valuable prognostic factor in patients with HCC	Gianelli et al. [59]
	Cervical carcinoma	Tumor tissue	Presence of mRNA for both MMP-2 and TIMP-2 is associated with poor survival	Davidson et al. [60]
		Serum	Medium level of TIMP-2 and MMP-2/TIMP-2 complex in serum is higher in healthy women, compared to those with a malignant tumor	Talvensaari-Mattila et al. [61]
	Oral cancer	Tumor tissue	A rate of MMP-2/TIMP-2 complex expression is better for characterization of MMP-2 activity	Shretsha et al. [62]
	Bladder cancer	Tumor tissue	The expression of MMP-2 and TIMP-2 was higher in invasive tumors, and was strongly associated with poorer survival	Kanayama et al [63].

MMP—metalloproteinase, TIMP—tissue inhibitor of metalloproteinase, BPH—benign prostatic hyperplasia, CSF—cerebrospinal fluid, HCC—hepatocellular carcinoma.

**Table 3 diagnostics-10-00547-t003:** Summary of selected studies of the role of MMP-7 in adult cancers.

Type of Cancer	Type of Sample	Result	Study
**Esophageal cancer**	Tumor tissue	Overexpression of MMP-7 is strictly connected with higher TNM stage, higher invasive grade, presence of lymph nodes metastasis	Miao et al. (meta-analysis) [71]
**Gastric cancer**	Tumor tissue	Overexpression of MMP-7 predicts presence of lymph node metastasis	Okayama et al. [72]
	Tumor tissue	MMP-7 levels are higher in gastric cancer with vascular invasion compared to the group without invasion	Yamashita et al. [73]
**Colorectal cancer**	Serum	MMP-7 is an independent prognostic factor for survival in advanced colorectal cancer, possibly even more accurate than LDH	Maurel et al. [74]
	Serum + Tumor tissue	Overexpression of MMP-7 in advanced cancer, compared to non-metastatic disease Higher level of MMP-7 in serum of stage III/IV patients, compared to patients in I/II stage disease	Polistena et al. [75]
**Pancreatic cancer**	Tumor tissue	MMP-7 positivity was correlated with poor prognosis	Yamamoto et al. [76]
	Tumor tissue	MMP-7 level was significantly increased in pancreatic cancer compared with healthy pancreas, and was related with reduced survival	Jones et al. [77]
	Plasma + Pancreatic juice	Median plasma MMP-7 levels were significantly higher in carcinoma, compared with chronic pancreatitis Combined MMP-7 and Ca 19-9 levels in plasma give positive predictive value of 100%	Kuhlmann et al. [78]
**Ovarian cancer**	Tumor tissue	Overexpression of MMP-7 in high-grade ovarian cancer and low malignant potential ovarian tumors; absence of MMP-7 expression in healthy ovary	Tanimoto et al. [79]
**NSCLC**	Tumor tissue	Overall survival significantly lower in patients with overexpression of MMP-7	Liu et al. [80]
**Bladder cancer**	Tumor tissue + serum	Overexpression of MMP-7 in tissue samples of metastatic bladder cancer, compared to those with localized disease High tissue expression levels are accompanied by elevated serum MMP-7 concentration	Szarvas et al. [81]

MMP—metalloproteinase, NSCLC—non-small cell lung cancer, LDH—lactate dehydrogenase, Ca 19-9—carbohydrate antigen 19-9.

**Table 4 diagnostics-10-00547-t004:** Summary of selected studies of MMP-2, MMP-7, and TIMP-2 role in pediatric cancers.

MMP/TIMP	Type of Cancer	Type of Sample	Result	Study
**MMP-2**	Acute lymphoblastic leukemia	bone marrow	In pediatric cases positive MMP-2 expression was in relationship with high-risk tumor group and T-cell immunophenotype	Kuittinen et al. [83]
	Acute myelogenous leukemia	adult cell-lines	Inhibiting the activity of MMP-2 significantly reduced invasiveness of leukemic cells	Sawicki et al. [84]
	Central nervous system tumors	tumor tissue	Distinctly positive expression of MMP-2 in anaplastic astrocytoma and glioblastoma compared with healthy brain and low-grade astrocytoma tissue	Gu et al. [85]
	Alveolar (ARMS) and embryonal (ERMS) rhabdomyosarcoma	ARMS and ERMS cell lines	ARMS cell line with increased expression of MMP-2 was more invasive compared to ERMS cell line with low levels of MMP-2.	Onisto et al. [86]
		ARMS and ERMS tumor samples	ARMS showed stronger MMP-2 expression, compared with ERMS type	Diomedi-Camassei et al. [87]
	Osteosarcoma	cell lines	The most invasive cell line contained the highest amounts of MMP-2	Bjorland et al. [88]
**MMP-2/TIMP-2**	Lymphomas	Plasma	Newly diagnosed lymphoma patients had higher levels of MMP-2/TIMP-2 complex and lower levels of TIMP-2, compared to healthy individuals Patients with highest plasma levels of MMP-2/TIMP-2 complex had a greater chance for relapse of the disease	Pennanen et al. [89]
	Neuroblastoma	Tumor tissue	Higher levels of MMP-2 mRNA and the ratio of MMP-2/TIMP-2 mRNA were observed in advanced stages and in patients who died from the progression of the disease	Ara et al. [90,91]
		Tumor tissue	Higher levels of MMP-2 in stage IV compared to stage I and II	Sugiura et al. [92]
		Tumor tissue	Expression of MMP-2 was increased in advanced stages of the disease	Ribatti et al. [93]
	Retinoblastoma (RB)	Tumor tissue	MMP-2 and TIMP-2 were overexpressed in invasive tumors	Adithi et al. [94]
		Cell lines	Higher expression of MMP-2 in poorly differentiated RB compared with moderately/well-differentiated RB Using inhibitors of MMP-2 and MMP-9 results in reduction in cell line migration ability, and significant decrease of cell viability	Webb et al. [95]
	Neuroblastoma RhabdomyosarcomaOsteosarcoma	Cell lines	In neuroblastoma in more aggressive cell lines, MMP-2 expression was twice as high as in less aggressive cells The highest level of MMP-2 was determined for osteosarcoma cell line In neuroblastoma, TIMP-2 was overexpressed in less aggressive line, compared with expression in more aggressive cell line	De Veas et al. [96]
**TIMP-2**	Lymphomas	Tumor tissue	TIMP-2 expression exclusively in lymphomas, none in reactive lymph nodes	Stolarska et al. [97]
	Hodgkin lymphoma	Tumor tissue	TIMP-2 expression correlate with systemic symptoms	Pennanen et al. [98]
**MMP-7**	Acute myelogenous leukemia	Cell lines	Only more aggressive cells expressed MMP-7 By inhibiting MMP-7 40% reduction of invasiveness was revealed	Lynch et al. [99]

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
