# Peer review of "Selected Matrix Metalloproteinases (MMP-2, MMP-7) and Their Inhibitor (TIMP-2) in Adult and Pediatric Cancer"

_diagnostics, 2020, doi:10.3390/diagnostics10080547_

Round 1

Reviewer 1 Report

This is updated , well ritteen careful article by the authors,

that according to this reviewer requires requires English revision.

Such detail that MMP-8 which is  a key collagenolytic MMP in the ethiopathogenesis of periodontal disease(Sorsa et al Ann Med-06 , Sorsa et al Periodontol-2000, 2016, Sorsa et al Diagnostics-20) can also contribute to developement of diabetes type II by processing insulin receptor a anon-matrix bioactive substrate and this can inhibited by doxycycline , an MMP-inhibitor, (Lauhio et al Eur J Clin Invest-16) could be amended a examples /detail into manuscript upon revision.

After these revisions the paper satrts to be suitable for publication-,

Author Response

Answers for the Reviewers

Review 1:

Thank you for the reviewer’ comments.

Comments and Suggestions for Authors

  • This is updated , well written, careful article by the authors, that according to this reviewer requires English revision.

Answer: the English revision has been made, as suggested.

  • Such detail that MMP-8 which is a key collagenolytic MMP in the ethiopathogenesis of periodontal disease(Sorsa et al Ann Med-06 , Sorsa et al Periodontol-2000, 2016, Sorsa et al Diagnostics-20) can also contribute to developement of diabetes type II by processing insulin receptor a anon-matrix bioactive substrate and this can inhibited by doxycycline , an MMP-inhibitor, (Lauhio et al Eur J Clin Invest-16) could be amended a examples /detail into manuscript upon revision.

Answer: We thank the Reviewer for the amended examples regarding the importance of MMP-8 in the disease iniciation and progression. We found those information really important to be added to the manuscript and the change is marked in blue.

The proposed articles are numbered in main text and in bibliography as: 14,15,16,17.

  1. Sorsa, T.; Tjäderhane, L.; Konttinen, Y.T.; Lauhio, A.; Salo, T.; Lee, H.M.; Golub, L.M.; Brown, D.L.; Mäntylä, P. Matrix metalloproteinases: contribution to pathogenesis, diagnosis and treatment of periodontal inflammation. Ann Med. 2006, 38(5), 306-21. DOI: 10.1080/07853890600800103.

  1. Sorsa, T.; Gursoy, U.K.; Nwhator, S.; Hernandez, M.; Tervahartiala, T.; Leppilahti, J.; Gursoy, M.; Könönen, E.; Emingil, G.; Pussinen, P.J.; Mäntylä, P. Analysis of matrix metalloproteinases, especially MMP-8, in gingival creviclular fluid, mouthrinse and saliva for monitoring periodontal diseases. Periodontol 2000 2016, 70(1), 142-63. DOI: 10.1111/prd.12101.

  1. Sorsa, T.; Alassiri, S.; Grigoriadis, A.; Räisänen, I.T.; Pärnänen, P.; Nwhator, S.O.; Gieselmann, D.R.; Sakellari, D. Active MMP-8 (aMMP-8) as a Grading and Staging Biomarker in the Periodontitis Classification. Diagnostics 2020 , 10(2), 61. DOI: 10.3390/diagnostics10020061.
  2. Lauhio, A.; Färkkilä, E.; Pietiläinen, K.H.; Åström, P.; Winkelmann, A.; Tervahartiala, T.; Pirilä, E.; Rissanen, A.; Kaprio, J; Sorsa, T.A.; Salo, T. Association of MMP-8 with obesity, smoking and insulin resistance. Eur J Clin Invest 2016, 46(9), 757-65. DOI: 10.1111/eci.12649.

Reviewer 2 Report

This is a great review of MMP-2, MMP-7, and TIMP-2 role in adult and pediatric cancer. 

Some minor suggestions to improve the manuscript. 

The title doesn't have to include "review of the involvement". 

It is not very clear why MMP-2, MMP-7, and TIMP-2 were selected among other pathological MMPs? Do they play a more significant role in pediatric cancers?

P. 2; line 66: If we count the numbers, it will be 23 MMPs, so why not replace more than 20 with 23?

P. 5; Figure 1. pdb ID (s)?

P. 5 use TIMP-3 or TIMP3 (one or another)

P. 5 line 177: ADAMs abbreviation?

P.5 line 201; cite recent papers that include metalloproteinase and their inhibitors role in cancer and developing therapeutics. 

P. 5 line 204; MMP-2 and MMP-7 are among the most important...

A few sentences in conclusion on comparison of these MMPs and TIMPs in adult and pediatric cancers? What is the difference and similarities?

Author Response

Review 2:

Thank you for the reviewer’ comments.

Answers for the Reviewer 1

  1. Comments and Suggestions for Authors

This is a great review of MMP-2, MMP-7, and TIMP-2 role in adult and pediatric cancer.

Some minor suggestions to improve the manuscript.

  • The title doesn't have to include "review of the involvement"

Answer: The authors corrected the title, as was suggested.  

  • It is not very clear why MMP-2, MMP-7, and TIMP-2 were selected among other pathological MMPs? Do they play a more significant role in pediatric cancers?

Answer: The levels of most metalloproteinases and their inhibitors may be disturbed in cancer patients. However, according to the available literature, the metalloproteinases selected by the authors have the most proven relationship specifically with the development of cancer, which is the main focus of the authors.

  • 2; line 66: If we count the numbers, it will be 23 MMPs, so why not replace more than 20 with 23?

Answer: We thank for the Reviewer’s comment. Indeed, there was a mistake, which was corrected. The change is marked in blue in the manuscript.

  • 5; Figure 1. pdb ID (s)?

Answer: the PDB for MMP-2 could be found here DOI: 10.2210/pdb1RTG/pdb while for TIMP-2: DOI: 10.2210/pdb1BR9/pdb. The IDs were added to the Figure 1 description.

  • 5 use TIMP-3 or TIMP3 (one or another)

Answer: Thank you for the kind suggestion. The TIMP-3 form has been chosen.

  • 5 line 177: ADAMs abbreviation?

Answer: The ADAMs abbreviation was explained once it appeared in the manuscript (please see P3 line 110).

  • 5 line 201; cite recent papers that include metalloproteinase and their inhibitors role in cancer and developing therapeutics.
    Answer: We thank for the Reviewer’s comment. The new papers have been added, the numbers in bibliography and in manuscript main text have been changed. They are numbered 37, 38 and 39:

  1. Raeeszadeh-Sarmazdeh, M.; Coban, M.; Sankaran, B.; Radisky, E. Engineering protein therapeutics for cancer based on the natural matrix metalloproteinase inhibitor TIMP‐1. Biochem Mol Biol 2020, 34(S1):1-1. DOI: 10.1096/fasebj.2020.34.s1.04889.

  1. Fischer, T.; Riedl, R. Inhibitory Antibodies Designed for Matrix Metalloproteinase Modulation. Molecules 2019, 24(12), 2265. DOI: 10.3390/molecules24122265.
  2. Zhong, Y.; Lu, Y.T.; Sun, Y.; Shi, Z.H.; Li, N.G.; Tang, Y.P.; Duan, J.A. Recent opportunities in matrix metalloproteinase inhibitor drug design for cancer. Expert Opin Drug Discov 2018; 13(1), 75-87. DOI: 10.1080/17460441.2018.1398732.

  • 5 line 204; MMP-2 and MMP-7 are among the most important...

Answer: The sentence has been changed.

  • A few sentences in conclusion on comparison of these MMPs and TIMPs in adult and pediatric cancers? What is the difference and similarities?

Answer: We thank the Reviewer for this suggestion. A few sentences were added to the manuscript in the summary section, marked in blue.

Round 2

Reviewer 1 Report

Concise, well done and written review.